# Structure of the exportin Xpo4 in complex with RanGTP and the hypusine-containing translation factor eIF5A

Metin Aksu[1], Sergei Trakhanov[1] & Dirk Görlich[1]

Xpo4 is a bidirectional nuclear transport receptor that mediates nuclear export of eIF5A and Smad3 as well as import of Sox2 and SRY. How Xpo4 recognizes such a variety of cargoes is as yet unknown. Here we present the crystal structure of the RanGTP•Xpo4•eIF5A export complex at 3.2 Å resolution. Xpo4 has a similar structure as CRM1, but the NES-binding site is occluded, and a new interaction site evolved that recognizes both globular domains of eIF5A. eIF5A contains hypusine, a unique amino acid with two positive charges, which is essential for cell viability and eIF5A function in translation. The hypusine docks into a deep, acidic pocket of Xpo4 and is thus a critical element of eIF5A's complex export signature. This further suggests that Xpo4 recognizes other cargoes differently, and illustrates how Xpo4 suppresses – in a chaperone-like manner – undesired interactions of eIF5A inside nuclei.

[1] Department of Cellular Logistics, Max Planck Institute for Biophysical Chemistry, Am Fassberg 11, 37077 Göttingen, Germany. Correspondence and requests for materials should be addressed to D.G. (email: goerlich@mpibpc.mpg.de).

The organization of eukaryotic cells is based on a division of labour between compartments as well as on inter-compartmental transport. Cell nuclei import all required proteins from the cytoplasm. The cytoplasm, in turn, relies on exported nuclear products such as transfer RNAs (tRNAs), messenger RNAs (mRNAs), as well as 40 and 60 S pre-ribosomal subunits (reviewed in ref. 1). The latter are exported in a still translation-incompetent state, but they gain full functionality soon after their arrival in the cytoplasm.

The nuclear membranes separate nucleus from cytoplasm, and force nucleocytoplasmic exchange to proceed through nuclear pore complexes (NPCs; reviewed in refs 2,3). The NPC permeability barrier controls this exchange and suppresses an intermixing of nuclear and cytoplasmic contents (reviewed in ref. 4). It grants free passage to small molecules, while restricting fluxes of larger species that approach or exceed a limit of ~5 nm in diameter or ~30 kDa in mass.

Shuttling nuclear transport receptors (NTRs) can overcome this size limit and carry larger cargoes across the barrier by means of facilitated translocation. Most NTRs have a HEAT repeat structure[5–8], bind RanGTP and draw energy from the RanGTPase system to transport cargoes against concentration gradients.

Importins recruit cargoes at very low RanGTP levels in the cytoplasm and translocate into the nucleus, where RanGTP displaces the cargo. The importin molecule then returns to the cytoplasm, where the RanGTP ligand is removed by a multi-step reaction that involves GTP-hydrolysis and loading of another cargo molecule. Exportins function the opposite way. They bind cargo cooperatively with RanGTP inside the nucleus and release cargo into the cytoplasm[9,10].

The exportin-functions known so far can be grouped into four categories: First, exportins perform 'biosynthetic export' of, for example, newly assembled pre-ribosomal subunits to the cytoplasm. Second, they act in regulatory circuits and keep, for example, certain transcription factors cytoplasmic, until an adequate stimulus inactivates the export signal and allows for nuclear accumulation. Third, exportins also assist nuclear import by recycling nuclear import adaptors, such as importin α or snurportin, back to the cytoplasm. Finally, exportins compensate for the imperfection of the NPC barrier and counteract the leakage of cytoplasmic components (such as actin or translation factors) into the nuclear compartment[11–14]. The latter appears crucial for an ordered course of gene expression, because the persistence of cytoplasmic translation activity inside nuclei would cause ribosomes to translate non-spliced, intron-containing mRNAs and hence lead to defective translation products.

Eight RanGTPase-driven exportins have been identified in mammals so far (reviewed in refs 15,16). How they select their cargoes has been a central question in the field of nucleocytoplasmic transport.

CRM1 (refs 10,17), also called exportin 1 or Xpo1, has the broadest cargo-spectrum and appears to export ~1,000 different protein species – either singly or as part of larger protein complexes[14]. It recognizes short (9–15 residues long) peptides called nuclear export signals or NESs (refs 18–20). Such NESs reside in disordered regions of the cargo, comprise 4–5 hydrophobic Φ-residues and are transplantable from one protein to another. CRM1 recognizes them by an NES-binding site with 5 hydrophobic pockets, into which the Φ-residues can dock[21–23].

Other exportins recognize their cargoes differently from CRM1, either because they bind RNA (in the cases of Xpo-t and Xpo5; refs 24,25), or because their cargoes must be exported in a 'special state'. The import adaptor importin α, for example, has to be exported with its nuclear localization signal (NLS)-binding site being securely locked[9,26], ensuring that NLS-containing proteins remain inside the nucleus. The responsible exportin (CAS/Xpo2) acts, therefore, not only as transporter, but as chaperone as well. For that, it recognizes not just a small peptide, but its cargo per se and in an auto-inhibited state.

CRM1 depletes most translation initiation and all termination factors from the nuclear compartment[12,14]; however, it does not appear to act on the elongation factors. This gap is filled by Xpo5, exporting tRNA•EF1A complexes (refs 12,13) and by Xpo4, the exportin for eIF5A (ref. 27).

eIF5A (refs 28,29), previously also referred to as IF-M2Bα or eIF4D, is universally conserved. It occurs not only in eukaryotes[28,30] and archaea[31], but also in eubacteria, where it is called EF-P (ref. 32). It is a translation elongation factor required for rapid synthesis of protein regions that contain consecutive proline residues[33–35]. Earlier literature assigned a multitude of other functions to eIF5A; retrospectively, however, it is reasonable to assume that these are secondary to the requirement of eIF5A for efficiently translating mRNAs that encode proteins with proline-rich motifs.

Eukaryotic and archaeal eIF5A comprises two domains[36,37], namely an N-terminal SH3-like domain and a C-terminal oligonucleotide-binding-fold domain. eIF5A contains a unique amino acid, namely an N-ε-(4-amino-2-hydroxybutyl)-modified lysine 50 (in human numbering) that is called hypusine[38–40]. The recently solved structure of yeast eIF5A bound to the E-site of cycloheximide-arrested ribosomes[41] revealed that hypusine contacts the CCA-end of the P-site tRNA. Hypusine is indispensible for eIF5A function and indeed for cell viability[42,43].

Fully hypusinated eIF5A is a cytoplasmic protein[44,45]. However, it is also rather small (17 kDa); it therefore enters nuclei rapidly and accumulates within nucleoli if not exported back by Xpo4 (ref. 27).

Xpo4 behaves like a tumour suppressor[44], and it is remarkable that Xpo4 exports not only eIF5A, but also other structurally unrelated cargoes, such as the transcription factor Smad3 (ref. 46). It even functions directly in nuclear import and carries Sox-type transcription factors[47] and possibly other proteins into the nucleus. No structural information on Xpo4 is available so far. It therefore remained unclear how Xpo4 can handle such a diversity of cargoes or indeed how it recognizes any of them.

In this paper, we report the first crucial step and present the x-ray structure of the RanGTP•Xpo4•eIF5A complex. Xpo4 has the shape of a toroid and is structurally closely related to CRM1, suggesting that Xpo4 might have evolved from CRM1. The NES-binding site of CRM1 is still recognizable, but it is occluded by an inserted NES-like element and a repositioned HEAT 11A helix. Instead, a newly evolved cargo-binding site is used, which is formed mainly by intra-repeat loops of HEATs 11-16, and which recognizes each of the two globular eIF5A domains. The interaction buries positively charged regions of eIF5A and the hypusine-containing loop in particular; it thereby also precludes off-target interactions of eIF5A within the nucleus. Xpo4 thus combines properties of an exportin with those of a compartment-specific chaperone.

## Results

**Structure determination.** For structural analysis, we assembled a mammalian RanGTP•Xpo4•eIF5A complex from bacterially expressed components. This included eIF5A (in vitro hypusinated by recombinant deoxyhypusine synthase[48,49] and deoxyhypusine hydroxylase[50]), as well as the GTPase-deficient RanQ69L mutant[51] comprising residues 5–180, which were resolved in previous RanGTP•NTR complexes[7,8,21,24–26,52].

Even though this complex was stable and monodisperse, it did not crystallize–apparently because of disordered loops and

termini: Indeed, the 14 N-terminal probably disordered residues of eIF5A (refs 36,37,53) not only turned out to be dispensable for complex formation (Fig. 1a), but also, their deletion allowed the RanGTP•Xpo4•eIF5A complex to crystallize in needle clusters. An additional *in situ* treatment with trypsin or chymotrypsin in the crystallization drops yielded triclinic crystals that diffracted to a resolution of 3.8 Å.

Limited proteolysis by trypsin or chymotrypsin did not compromise the integrity of the heterotrimeric complex (Fig. 1b). It also left eIF5A and Ran intact, but cleaved Xpo4 within two poorly conserved loops comprising residues 241–260 and 931–948, respectively (Fig. 1c and Supplementary Fig. 1). Deleting these loops led to an Xpo4 mutant that still bound Ran and eIF5A like the wild-type exportin (Fig. 1d,e), but became trypsin-resistant (Fig. 1f) and yielded triclinic export complex crystals even without protease addition.

The structure of this ternary export complex was solved by a combination of molecular replacement (MR) using Ran (PDB ID 3GJX; ref. 21) and eIF5A (PDB ID 3CPF; ref. 37) as search models and single-wavelength anomalous dispersion (SAD) phasing with

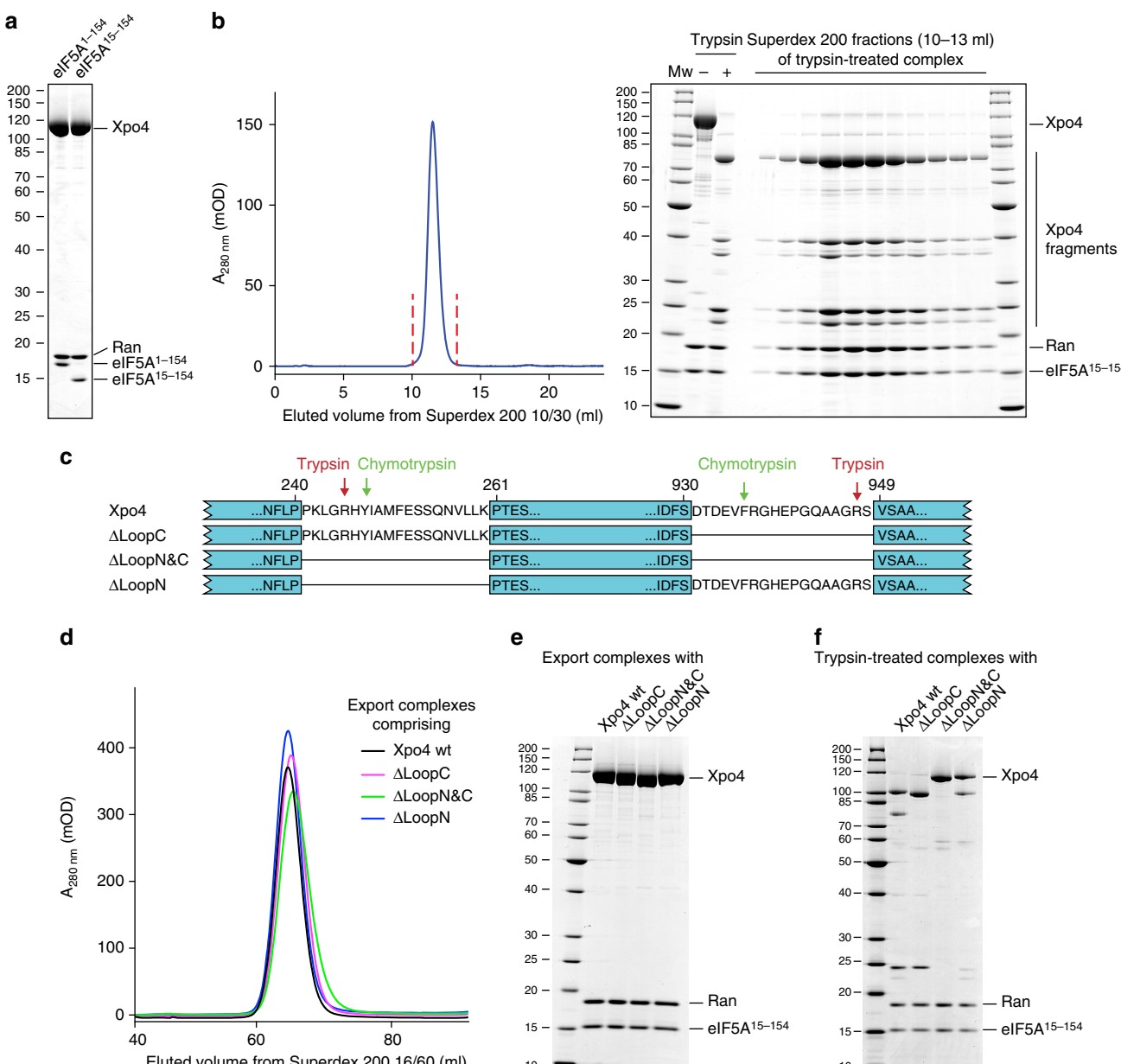

**Figure 1 | Improvement of Xpo4 export complex crystallization by removal of flexible parts.** (**a**) Export complexes comprising either full-length or truncated eIF5A were formed with Xpo4 and ZZ-bdNEDD8-tagged RanGTP. After immobilization via tagged Ran, bound proteins were eluted by bdNEDP1 protease[59], and analysed by SDS–polyacrylamide gel electrophoresis (SDS–PAGE) and Coomassie-staining. (**b**) 180 μg of the export complex was incubated with trypsin (500:1 w/w) for 90 min at 22 °C. The reaction was stopped by 5 mM PMSF and EDTA. Note that the resulting proteolyzed complex remained intact when analysed by size exclusion chromatography (SEC) on a Superdex 200 10/30 column. Experiments with chymotrypsin gave similar results. (**c**) Illustration of Xpo4 mutants. Protease cleavage sites were identified by mass spectrometry. (**d**) Overlayed size exclusion chromatograms of the export complexes derived from wild-type Xpo4 or Xpo4 loop deletions. (**e**) Peak fractions from **d** were pooled, concentrated and analysed by SDS–PAGE followed by Coomassie-staining. (**f**) The export complexes from **e** were incubated with trypsin (1,000:1 w/w) for 1 h at 22 °C and analysed by SDS–PAGE and Coomassie-staining. The ΔLoopN&C Xpo4 variant was trypsin-resistant.

**Table 1 | Data collection and refinement statistics for RanGTP•Xpo4•eIF5A complex (MR-SAD).**

|  | Native | SeMet |
|---|---|---|
| *Data collection* |  |  |
| Space group | $P3_121$ | $P3_121$ |
| Cell dimensions |  |  |
| $a, b, c$ (Å) | 98.62, 98.62, 726.86 | 98.48, 98.48, 725.68 |
| $\alpha, \beta, \gamma$ (°) | 90.00, 90.00, 120.00 | 90.00, 90.00, 120.00 |
| Resolution (Å) | 49.41 – 3.20 | 49.33 – 3.48 |
|  | (3.31 – 3.20)* | (3.61 – 3.48) |
| $R_{sym}$ | 0.08 (0.95) | 0.15 (1.96) |
| $I/\sigma I$ | 22.42 (2.26) | 18.00 (1.40) |
| Completeness (%) | 99.91 (99.79) | 99.70 (97.10) |
| Redundancy | 9.90 (10.20) | 18.70 (15.10) |
|  |  |  |
| *Refinement* |  |  |
| Resolution (Å) | 49.41 – 3.20 | – |
| No. reflections | 69,887 | – |
| $R_{work}/R_{free}$ (%) | 21.5/26.8 | – |
| No. atoms |  |  |
| Protein | 20,622 | – |
| Ligand/ion | 66 | – |
| Water | 0 | – |
| *B*-factors |  |  |
| Protein | 118.88 | – |
| Ligand/ion | 75.92 | – |
| Water | – | – |
| r.m.s. deviations |  |  |
| Bond lengths (Å) | 0.006 | – |
| Bond angles (°) | 0.57 | – |

*Values in parentheses are for highest-resolution shell.*

crystals that contained selenomethionine-substituted Xpo4. The final model was refined against a native data set to 3.2 Å resolution, an $R_{work}$ of 21.5% and an $R_{free}$ of 26.8% (Table 1). Supplementary Fig. 2 depicts two details of the structure to exemplify the quality of the electron density.

The asymmetric unit comprises two ternary complexes of very similar structure (r.m.s.d. of ~1 Å). The electron density of complex 1 (corresponding to chains B, C and F in pdb 5DLQ) is, however, slightly better defined. The final model includes residues 7–179 of Ran and 15–152 of eIF5A. We modelled 1,052 of 1,113 residues of Xpo4, with a few residues missing from N- and C-terminus as well as from several loop regions between the HEAT repeats.

**Overall structure of Xpo4.** Xpo4, as all members of the importin β family, is an α-helical protein built of consecutive HEAT repeats (Fig. 2b,c). HEAT repeats are ~40 amino acid motifs, which consist of two α-helices (A and B) that pack in an anti-parallel orientation against each other[54]. Individual HEAT repeat units, in turn, pack side by side with a clockwise rotation to form a superhelical structure, whereby the A helices form the outer convex surface and the B helices the inner (concave) one.

Xpo4 consists of 19 canonical HEAT repeats and three additional α-helices (termed HEAT 20) that cap the superhelix at the very C-terminus (Fig. 2b,c). The superhelical arrangement of Xpo4 is interrupted by three anticlockwise kinks (between HEATs 3 and 4, HEATs 9 and 10, and HEATs 13 and 14) that bend the superhelical structure into a toroid-like shape, whereby HEAT 20 touches the loop between HEATs 2 and 3. This architecture is very similar to the cargo-bound form of CRM1 (refs 21,22; see below, Fig. 4), a difference being that Xpo4 lacks the most C-terminal helix of CRM1. In turn, Xpo4 contains a

number of insertions either between the A and B helices of the same HEAT repeat (intra-repeat inserts) or between successive HEAT repeats (inter-repeat inserts). As mentioned already, two poorly conserved insertions had to be deleted to obtain a crystallizable complex. The conserved ones will be discussed below.

**RanGTP recognition by Xpo4.** Xpo4 wraps with four distinct Ran-interaction sites around Ran and thereby buries a large surface area of ~1,920 Å$^2$ (Figs 2a,c and 3). The first Ran-binding site is formed by HEATs 1–3 (Fig. 3) and represents the region that is most conserved between importin β superfamily members[5,55]. This N-terminal site contacts Ran's switch II region (residues 65–80) as well as α-helix 3 (Fig. 3). The interaction occurs mainly via hydrophobic contacts, which is very similar to what has been seen with other NTRs before[7,21,24–26,52].

HEATs 7 and 8 form a second, acidic interaction site with Asp395, Asp396 and Glu401 contacting a basic region of Ran (His139, Arg140, Gln145, Trp163 and Arg166; Fig. 3). Overall, this interaction resembles the interaction between Ran and the conserved acidic insertion of importin β and transportin[7,8].

The third interaction site of Xpo4 is formed by the acidic loop within HEAT 9, which contacts the guanine-binding loops of Ran (Fig. 3). CRM1 contacts the same region of Ran with the same acidic loop as well. There is, however, a notable difference: while CRM1-Ran contacts are mostly electrostatic[21], Xpo4 makes hydrophobic contacts, mainly through Leu466. The exportins CAS, Xpo-t and Xpo5 also contact this Ran region, but through loops within C-terminal HEAT-repeats[24–26]. The last Ran-binding interface of Xpo4 involves its C-terminal HEAT repeats 16 and 17, which bind the switch I region of Ran (residues 30–47; Fig. 3). Other exportins (except for Xpo5) contact the switch I region of Ran also by the C-terminal HEAT repeats.

The structure of Ran in the export complex is almost identical (with an r.m.s.d of ~0.5 Å) to that in other NTR–RanGTP complexes. Although the details of the interactions of Xpo4 with Ran vary at certain regions, the overall recognition mechanism is similar to that seen in other exportins. Xpo4 contacts switches I and II of Ran, hence directly sensing its nucleotide-bound state. The different conformations of these regions in GDP-bound Ran[56,57] would cause extensive clashes with HEATs 1–3 in the export complex and thus make RanGDP incompatible with Xpo4-binding.

Essential steps of the export cycle are cytoplasmic cargo-release and hydrolysis of the Ran-bound GTP molecule[9,10,27]. Yet, RanGAP cannot trigger GTP-hydrolysis directly, because Xpo4 masks the RanGAP-binding site of Ran entirely (see Supplementary Fig. 3, and refs 27,58). Instead, the GAP co-activator RanBP1 probably interacts initially with Ran's C-terminus (that is dispensable for exportin-binding), and while engaging into a tighter complex, it will displace Ran and cargo from the exportin. RanGAP can then attack the transiently released RanBP1•RanGTP complex, and the resulting GTP-hydrolysis makes the dissociation irreversible. The ligand-free Xpo4 molecule can then either load an import cargo or directly return to the nucleus for another round of export.

**Cooperativity between RanGTP and cargo-binding to Xpo4.** (Nuclear) RanGTP increases the affinity of exportins for their cargoes, and there are two possibilities of how this may happen. First, Ran and cargo form a common interaction interface and the released binding energy promotes the exportin-binding of the other ligand[24–26]. This principle applies to Xpo-t (exporting tRNAs), Xpo5 (exporting tRNAs and pre-microRNAs) and Xpo2 (exporting importin α). However, Ran and cargo do not contact

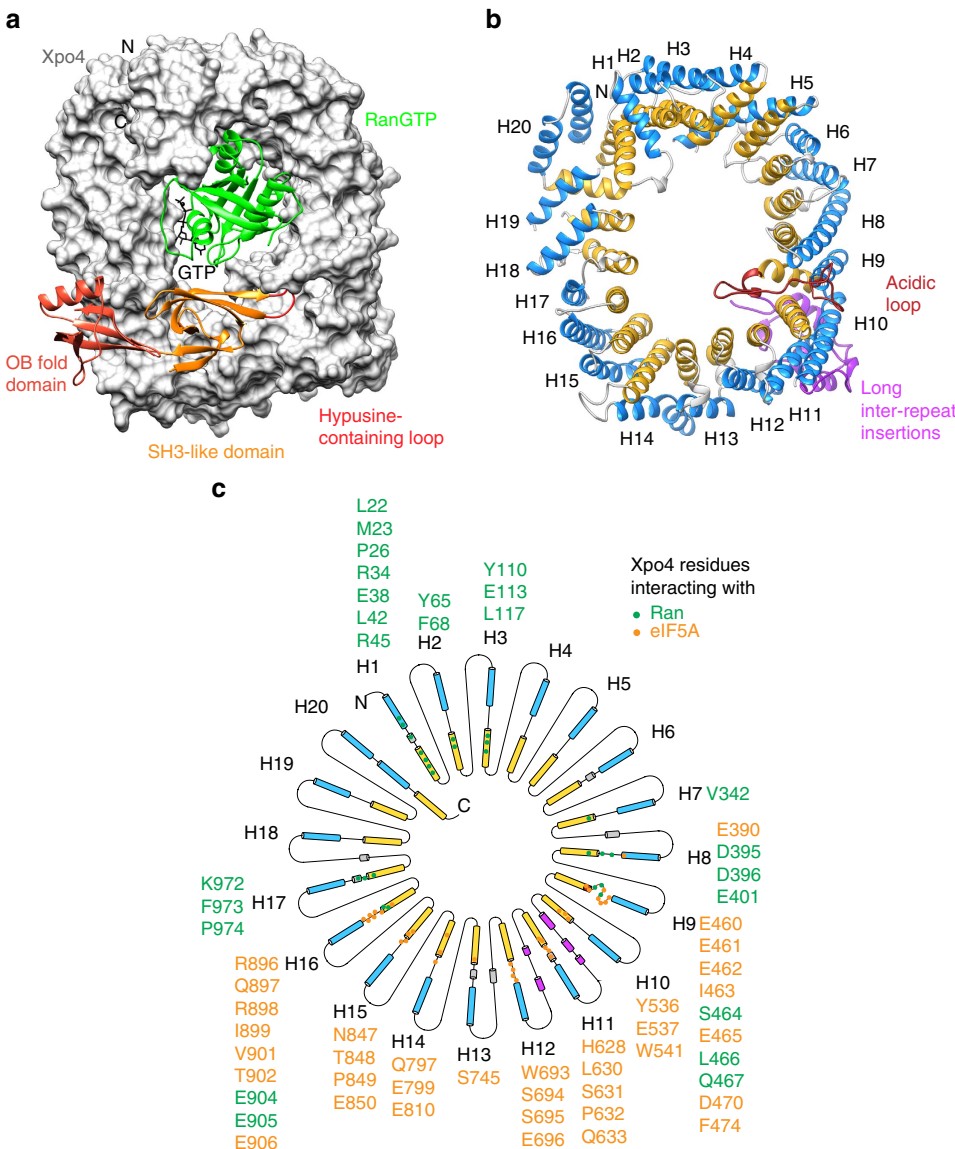

**Figure 2 | Structure of the RanGTP•Xpo4•eIF5A export complex and HEAT repeat organization of Xpo4.** (**a**) Xpo4 (grey) is shown as surface representation, while Ran (green) and eIF5A are shown as ribbon representation. Different regions of eIF5A are coloured and labelled accordingly. GTP is shown as black sticks. (**b**) Xpo4 in the export complex is shown in a ribbon representation (RanGTP and eIF5A are removed for clarity). Colour scheme: A helices of the HEAT repeats, blue; B helices, yellow; long inter-repeat insertions, dark pink; the acidic loop, brown. (**c**) Schematic representation of the Xpo4 secondary structure. Colouring is as in **b**. Green and orange dots represent the Xpo4 residues interacting with RanGTP and eIF5A, respectively.

each other in a CRM1-export complex[21]. Here, it appears that the nuclear exportin conformation is 'spring-loaded' and that cooperativity is achieved by each of the two ligands stabilizing the same high-energy conformation of CRM1.

In the case of the Xpo4 export complex, we found just a very small Ran-eIF5A interface, comprising no more than a single salt bridge between R29[Ran] and E42[eIF5A] (Supplementary Fig. 4a). Abolishing this salt bridge by an E42A[eIF5A] mutation had no detectable effect on export complex formation (see below, Fig. 7c). This suggests that Xpo4, like CRM1, relies on an allosteric mechanism for RanGTPase-driven cargo-loading and unloading.

**Divergence of Xpo4 from CRM1.** A plausible scenario is that Xpo4 arose from a duplication of the CRM1 gene early in eukaryotic evolution, and then diverged into an orthogonal cargo specificity, which includes that CRM1-specific NESs are no longer

accepted. The remnants of the NES-binding site[21,22] in CRM1 (formed by HEATs 11 and 12) are still recognizable in Xpo4, and helices 11B, 12A and 12B still nearly perfectly align (Fig. 4). However, a repositioning of the HEAT 11A helix closes one side of the hydrophobic cleft, while the remainder of the cleft is blocked by inserts between the canonical HEAT helices. One of these inserted regions actually resembles in structure and position a PKI-type NES (ref. 23).

**Interactions of Xpo4 with its export cargo eIF5A.** Xpo4 neither binds a small NES peptide to a hydrophobic cleft, nor does Xpo4 enwrap its cargo. Instead, eIF5A sits on an unprecedented binding site provided by HEATs 8–16, and in particular by the intra-repeat loops of HEATs 9, 11, 12 and 14–16 (Fig. 2a,c). Consistent with earlier mapping data[27], Xpo4 interacts extensively with both domains of eIF5A (refs 36,37), namely with the N-terminal SH3-like domain as well as with the

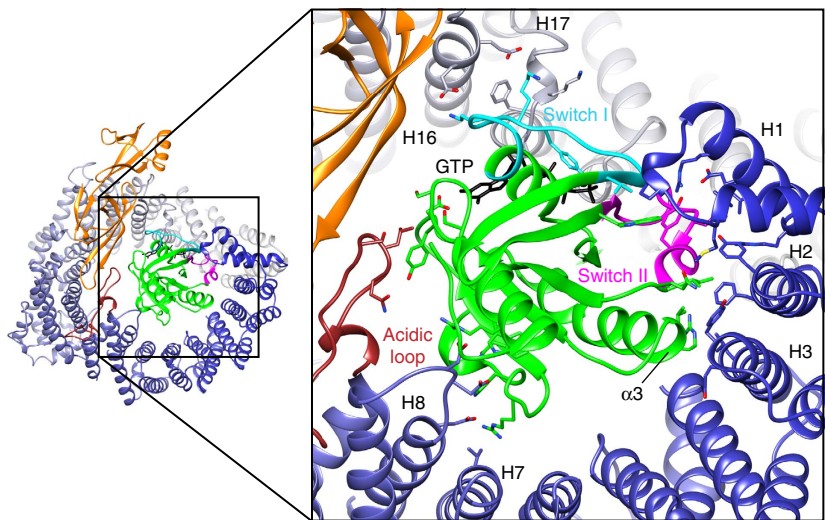

**Figure 3 | RanGTP recognition by Xpo4.** The export complex is shown as a ribbon representation. On the left, Xpo4 is coloured in a gradient from blue (N-terminus) to grey (C-terminus); the acidic loop is shown in brown. eIF5A is coloured in orange and Ran in green. Switch I and II regions of Ran are shown as cyan and pink, respectively. GTP (black) is shown as sticks. The close-up on the right illustrates Xpo4-Ran interactions.

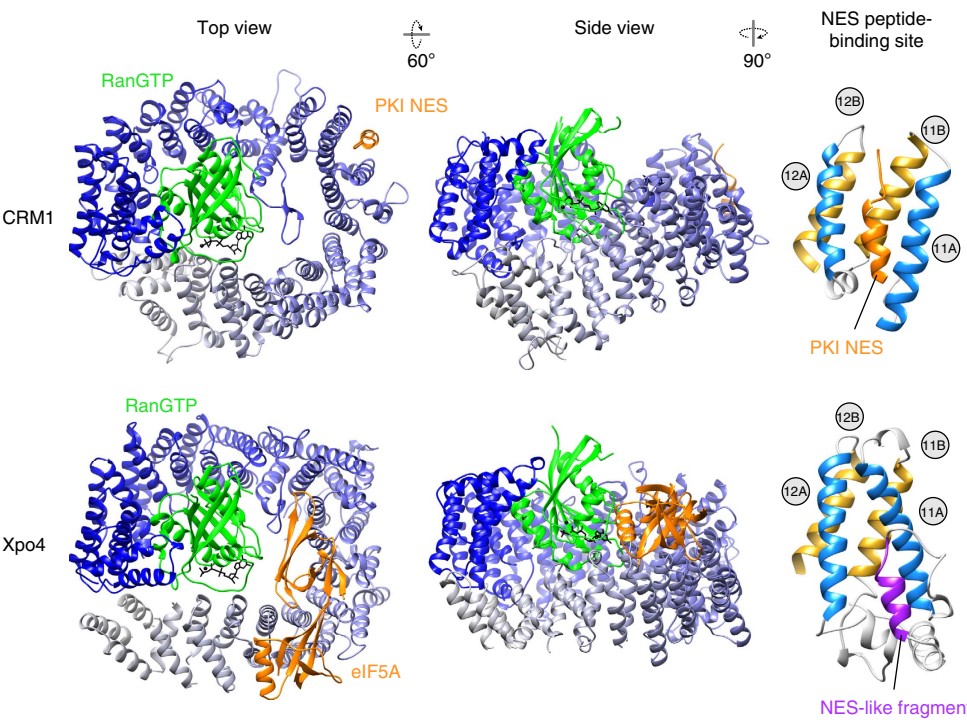

**Figure 4 | Comparison of the ligand-bound structures of CRM1 and Xpo4.** Export complexes are aligned with respect to Ran, illustrated in ribbon representations, and each shown in two different orientations (side and top views). Exportins are coloured in gradients from blue (N-terminus) to grey (C-terminus). The respective cargoes are shown in orange, Ran in green. Right, NES-binding site of CRM1 and the corresponding region of Xpo4 are shown. A and B helices of the HEAT repeats are coloured in blue and yellow, respectively. The PKI NES is coloured in orange, the NES-like fragment in Xpo4 in purple.

C-terminal oligonucleotide-binding (OB)-fold domain (Figs 2a,5a and Supplementary Fig. 4).

The interaction buries a very large surface area of 2169 Å$^2$. Xpo4 shields many of the positively charged and highly conserved residues that constitute eIF5A's 25S RNA- and tRNA-binding interface (Fig. 5a–c; ref. 41). This qualifies Xpo4 as a nucleus-specific inhibitor (or chaperone) of eIF5A and as an antagonist towards off-target interactions of eIF5A, for example, with nuclear RNA species (discussed below). The recognition of very conserved eIF5A features is also evident from our observation

that mammalian Xpo4 binds yeast (*S. cerevisiae*) eIF5A perfectly well (M.A., A. Rodriguez, D.G., unpublished data). It appears that Xpo4 has adapted to pre-existing eIF5A features.

The C-terminal oligonucleotide-binding-fold domain of eIF5A forms the smaller binding interface, but interacts extensively with Xpo4. It sits on a concave surface made up of the intra-repeat loops of HEATs 14–16 (Supplementary Fig. 4b). The intra-repeat-loop of HEAT 16 runs in opposite direction to β-strand 11 and makes backbone hydrogen bonds as well as hydrophobic contacts via I149$^{eIF5A}$. Several residues of the intra-repeat loops of HEATs

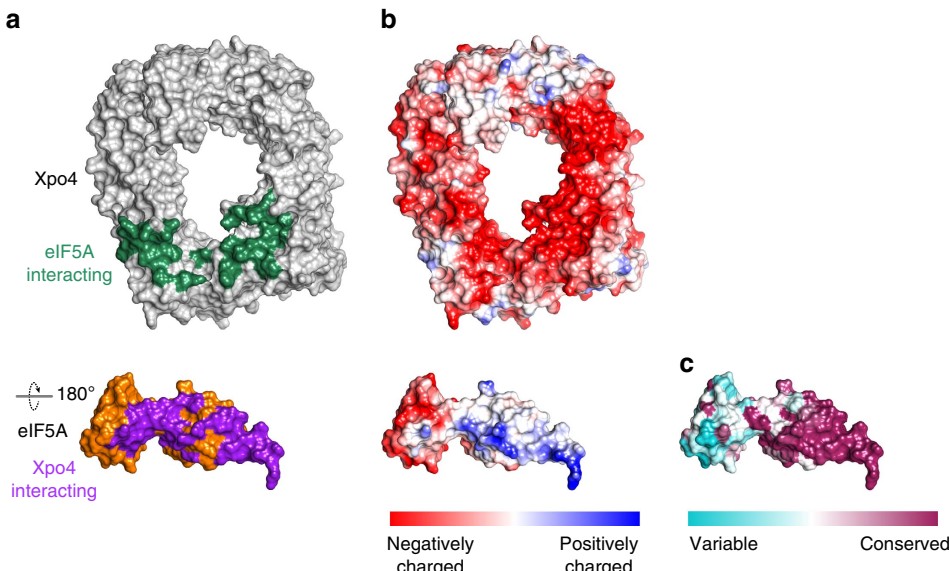

**Figure 5 | eIF5A recognition by Xpo4.** Xpo4 and eIF5A are rendered as surface representations, with RanGTP being removed for clarity. Rotation of eIF5A is indicated. (**a**) Xpo4 is coloured in grey and eIF5A in orange. Interaction surface of Xpo4 on eIF5A is shown in dark green, whereas that of eIF5A on Xpo4 in dark pink. (**b**) Xpo4 and eIF5A are coloured according to electrostatic potential with a colour gradient from red (negatively charged) to blue (positively charged). (**c**) eIF5A is coloured according to conservation with a colour gradient from cyan (variable) to maroon (conserved). Conservation was based on 50 sequences ranging from animals (human), fungi, plants to protozoans (*Leishmania mexicana*).

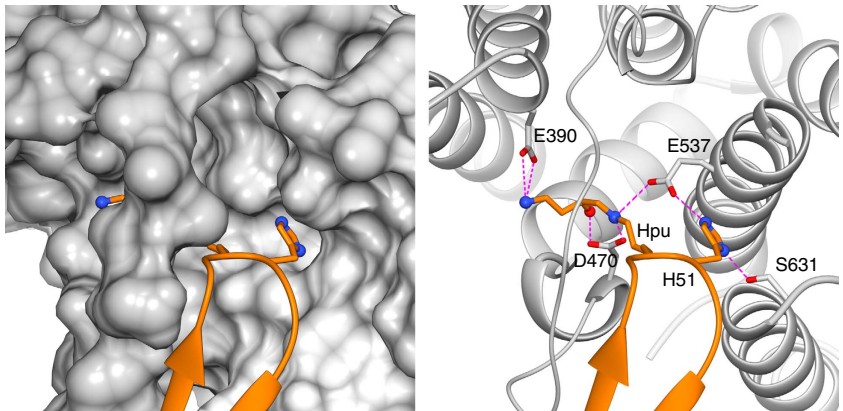

**Figure 6 | Docking of the hypusine-containing loop into the acidic pocket of Xpo4.** eIF5A is shown as orange ribbon, while the hypusine (Hpu) and histidine (H51) are shown as sticks. Xpo4 is coloured in grey and depicted as surface on the left and as ribbon on the right. The Xpo4 residues that interact with hypusine and H51 are shown as sticks. Nitrogens are shown as blue spheres, oxygens in red.

14 and 15 approach eIF5A and interact via polar contacts with β-strands 7 and 10 (β7 and β10, respectively).

The N-terminal SH3-like domain of eIF5A constitutes the larger interaction interface and contacts numerous residues on HEATs 8–16 of Xpo4 (Supplementary Fig. 4a). Most of these interactions involve salt bridges and hydrogen bonds. The 'basic tip of eIF5A', comprising β-strand 3, the hypusine-containing loop and β-strand 4, forms the centre of the interactions (Supplementary Fig. 4a). The acidic loop of Xpo4 forms an antiparallel inter-chain β-sheet with β-strand 3 of eIF5A and thereby locks the position of the basic tip. In addition, E462$^{Xpo4}$ and E465$^{Xpo4}$ of the acidic loop contact T45$^{eIF5A}$, T48$^{eIF5A}$ and K55$^{eIF5A}$ on the basic tip and further stabilize this interaction.

The perhaps most striking interaction was observed for the hypusine (Hpu50) and the neighbouring histidine (H51) (Fig. 6). The hypusine side chain bends in an L-shape and hooks into a crooked pocket that is formed by the acidic loop of Xpo4 and

HEAT 10. E390$^{Xpo4}$ engages in a salt bridge with the terminal κ-amino group of hypusine, E537$^{Xpo4}$ forms salt bridges with the ε-amine group of Hpu50$^{eIF5A}$ and with H51$^{eIF5A}$, while D470$^{Xpo4}$ makes another salt bridge with the ε-amino group and a hydrogen bond with the hypusine hydroxyl group. S631$^{Xpo4}$ stabilizes H51$^{eIF5A}$ further.

Consequently, we mutated the above-mentioned Xpo4 residues, and analysed the resulting mutants. None of the mutations impaired the Ran•Xpo4 interaction (Fig. 7a), verifying that folding was not affected. However, even the very subtle S631A$^{Xpo4}$ or the isosteric E537Q$^{Xpo4}$ and D470N$^{Xpo4}$ mutations abolished eIF5A binding (Fig. 7a).

Nuclear export assays provided the same striking result: in the absence of Xpo4, fluorescent eIF5A accumulated (weakly) in the nucleoplasm and strongly within nucleoli (Fig. 7b). Addition of wild-type Xpo4 led to complete loss of the nuclear/nucleolar signal. The S631A, E537Q and D470N Xpo4 mutants, however, had no such effect.

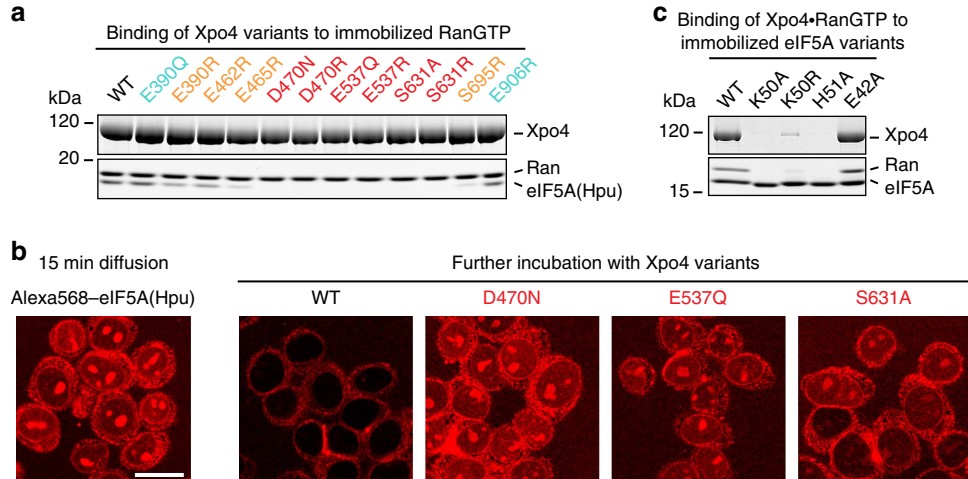

**Figure 7 | Xpo4 residues contacting hypusine and H51$^{eIF5A}$ are crucial for eIF5A binding and export. (a)** 1 µM Xpo4 wild type or mutants were incubated with 0.75 µM ZZ-bdNEDD8-tagged RanGTP and 1.25 µM hypusinated eIF5A in a 100 mM NaCl buffer. Formed complexes were retrieved via tagged Ran, eluted by (the tag-cleaving) bdNEDP1 protease, and analysed by SDS–polyacrylamide gel electrophoresis (SDS–PAGE) and Coomassie-staining. Uncropped gels are shown in Supplementary Fig. 5. **(b)** Alexa568-labelled hypusinated eIF5A (2 µM) was allowed to diffuse into the nuclei of digitonin-permeabilized HeLa cells[74] in the presence of an energy-regenerating system and an NTR-depleted extract[73] prepared from unfertilized *Xenopus* eggs. The mixture was split 15 min later and indicated Xpo4 variants (2 µM) were added. After 30 min, eIF5A distributions were recorded by confocal fluorescence microscopy. Scale bar, 20 µm. **(c)** The binding assay was performed as in **a**, but 1 µM untagged RanGTP and Xpo4 were incubated with 0.75 µM ZZ-bdSUMO tagged (non-hypusinated) wild-type or mutant eIF5A in a 50 mM NaCl buffer, and bdSENP1 was used for elution.

**Why is a special exportin needed to keep eIF5A cytoplasmic?**
CRM1 mediates nuclear export of ~1,000 different proteins, many of which are factors that have to be kept exclusively cytoplasmic[14]. Why then is eIF5A not amongst the CRM1 substrates? Why have eukaryotes evolved a specialized exportin for confining eIF5A to the cytoplasm? Why can't cells prevent a nuclear entry of eIF5A in the first place? And why should nuclear accumulation of eIF5A be a problem?

Nuclear entry of eIF5A not only leads to its depletion from the cytoplasm, but also, it is evident that it binds strongly to nucleolar structures. Such off-target interactions might disturb nucleolar functions, for example, the assembly process of ribosomal subunits. Furthermore, as nearly all translation factors are actively depleted from the nuclear interior, we can also see eIF5A export as part of the cell's effort to keep nuclei free of translation activity.

eIF5A has an invariably small molecular mass of only 17 kDa and can thus cross the permeability barrier of NPC rather rapidly even without the help of any importin. An increase in size would have been a straightforward way of slowing down such passive nuclear entry. Yet, a fusion to a sufficiently large 'cytoplasmic anchor' was not an evolutionary successful solution, perhaps because the present (small-sized) forms of eIF5A represent the optimum for rapid and accurate translation, and because even slight reductions in translation efficiency would decrease the competitive fitness of the organism.

An explanation for its CRM1-independent export is that eIF5A must be transported in a state that precludes interactions with nuclear off-targets. Otherwise, nuclear retention might dominate, or -even worse- nuclear interaction partners of eIF5A might get exported as well. Xpo4 appears indeed highly optimized to cleanly 'extract' eIF5A from the nuclear interior. It covers the most conserved and the most positively charged (RNA-interacting) regions of eIF5A and should thus be considered as an eIF5A chaperone. As an exportin, however, it can suppress nuclear eIF5A interactions not only stoichiometrically, but also by means of multi-round export in a 'catalytic' manner.

**Outlook**. Another fascinating facet of Xpo4 is that it carries not only eIF5A, but fully unrelated cargoes as well. We already know of Smad3 as an additional export[46] and of Sox-type transcription factors as import loads[47]. Given that the full cargo-spectrum of Xpo4 has not yet been explored, it is even possible that Xpo4 actually transports a far larger number. From a structural perspective, it will then be interesting to see how different the other cargo-binding sites are, how RanGTP displaces the import substrates and in how far distinct types of cargoes influence each other's transport behaviour.

## Methods
**Protein expression and purification.** All Xpo4 variants were expressed as N-terminal His14-bdSUMO fusions in *E.coli* Top10 F' at 21 °C for ~16 h. The cells were resuspended in buffer A (50 mM Tris/HCl pH 7.7, 500 mM NaCl, 2 mM Mg(OAc)₂ and 2 mM dithiothreitol (DTT)) supplemented with 5% glycerol and lysed by sonication on ice. Lysates were cleared by ultracentrifugation, and the protein was bound to a Ni (II) chelate matrix in the presence of 15 mM imidazole/HCl pH 7.7. The column was first washed with buffer A and then with buffer B (50 mM Tris/HCl pH 7.7, 150 mM NaCl, 2 mM Mg(OAc)₂ and 2 mM DTT) supplemented with 25 mM imidazole, and Xpo4 finally eluted by 250 nM of the tag-cleaving bdSENP1 protease[59].

The eluate was subjected to a Superdex 200 16/60 gel filtration column (GE Healthcare) equilibrated with buffer B. For crystallization, Xpo4 was further purified via anion exchange chromatography (Mono Q HR 5/5 column, GE Healthcare). For phasing, selenomethionine-substituted Xpo4 was expressed in BL21 cells grown in minimal medium supplemented with lysine, phenylalanine, threonine, isoleucine, leucine, valine and selenomethionine. Selenomethionine-labelled Xpo4 was purified using the same protocol as unmodified Xpo4. The DTT concentration was increased to 5 mM during the purification.

RanQ69L (aa 5–180) was expressed as an N-terminal His14-ZZ-bdSUMO fusion in *E.coli* Top10 F' cells at 21 °C for ~16 h. The cells were lysed in buffer C (50 mM HEPES/KOH pH 8.2, 500 mM NaCl, 2 mM MgCl₂ and 2 mM DTT). Ran was bound in the presence of 20 mM imidazole from the cleared lysate to a Ni (II) chelate matrix and eluted with buffer C containing 500 mM imidazole. To assess the nucleotide-bound state of Ran, a protein sample was boiled to release the nucleotide, and the nucleotide pattern was analysed by anion exchange chromatography and compared with that of a standard GDP–GTP mixture. For binding assays, Ran was expressed as an N-terminal His14-bdSUMO-ZZ-bdNEDD8 fusion, bound to Ni (II) chelate matrix and eluted with bdSENP1 protease as described above.

All eIF5A variants were expressed as N-terminal His14-ZZ-bdSUMO fusions in *E.coli* NEB Express cells as described for Ran, but buffer A was used throughout the purification.

Deoxyhypusine synthase and deoxyhypusine hydroxylase were expressed as N-terminal His10-GFP-Tev and His21-Tev fusions, respectively, as described for eIF5A variants. The two enzymes were purified by Ni(II) chelate chromatography and used in tagged form, so that they could be removed after the reaction.

**In vitro eIF5A hypusination.** Deoxyhypusination and hydroxylation reactions were adapted from published protocols[27,60] and optimized for higher modification efficiency. His14-ZZ-bdSUMO-tagged eIF5A (160 μM) was incubated with 4 μM deoxyhypusine synthase, 20 μM deoxyhypusine hydroxylase, 2 mM NAD, 2.5 mM spermidine, 5 mM DTT, 50 mM Tris/HCl pH 7.5 for 16 h at room temperature. Afterwards, another 20 μM deoxyhypusine hydroxylase was added and the incubation continued for 4 h at 37 °C.

eIF5A was separated from the enzymes by binding to an anti-ZZ affinity matrix. Bound protein was eluted by bdSENP1 protease in 15 mM Tris/HCl pH 7.0, 150 mM NaCl and 2 mM DTT.

The eluate was diluted to 50 mM NaCl with 15 mM Tris/HCl pH 7.0 and bound to a HiTrap SP HP 5 ml column (GE Healthcare) equilibrated with 15 mM Tris/HCl pH 7.0, 20 mM NaCl and 2 mM DTT. Still non-modified eIF5A was eluted with 50 mM Tris/HCl pH 7.0, 300 mM NaCl and 2 mM DTT. Elution of deoxyhypusinated and hypusinated eIF5A was with 50 mM Tris/HCl pH 7.0, 600 mM NaCl and 2 mM DTT.

To remove deoxyhypusinated eIF5A, we exploited the observation that the first modification step is reversible as long as hydroxylation has not yet occured[61]. For enzymatic removal of deoxyhypusine, 100 μM modified eIF5A and 2.5 μM His10 tagged deoxyhypusine synthase were incubated with 2 mM NAD, 2 mM 1,3-diaminopropane, 2 mM DTT, 200 mM glycine pH 9.0 for 4 h at 37 °C. The enzyme was separated from eIF5A by Ni (II) chelate resin. The reversion of the deoxyhypusination reaction converts deoxyhypusine to lysine while having no effect on hypusinated eIF5A. eIF5A(Hpu) was separated from eIF5A(Lys) by cation exchange chromatography as detailed above.

**Reconstitution and crystallization of the ternary complex.** For complex formation, Xpo4 variants and His14-ZZ-bdSUMO-tagged RanGTP were mixed with a 1.3-molar excess of hypusine-containing eIF5A. After 1 h incubation on ice, the buffer was exchanged to buffer D (15 mM Tris/HCl pH 7.7, 48 mM NaCl, 2 mM Mg(OAc)$_2$ and 2 mM DTT) and the sample was further incubated on ice for 3 h. The complex was immobilized on anti-ZZ beads via tagged RanGTP. Unbound proteins were removed; RanGTP and the bound proteins were eluted by bdSENP1 protease in buffer D. The eluate was subjected to a Superdex 200 16/60 gel filtration column (GE Healthcare) equilibrated with 15 mM Tris/HCl pH 7.7, 18 mM NaCl, 2 mM Mg(OAc)$_2$ and 2 mM DTT. The purified complex was concentrated to 12 mg ml$^{-1}$.

Diffracting quality crystals were obtained at 20 °C using microseeding hanging drops by mixing 1 μl of the protein solution with 1 μl reservoir solution containing 100 mM MES pH 6.26 and 8–11% PEG 400. Crystals were slowly transferred to a cryo-protectant solution (100 mM MES pH 6.26, 26% PEG 400, 15% Glycerol) and flash-frozen in liquid nitrogen.

**Structure determination and analysis.** All diffraction data were collected at the beamline X10SA (Swiss Light Source, Villigen, Switzerland). All data sets were indexed, integrated and scaled with XDS (ref. 62). For phasing, 56 selenium sites (out of 66) were located by SHELXD (ref. 63). Initial phases were obtained by molecular replacement with PHASER (ref. 64) using Ran (PDB ID 3GJX, ref. 21) and eIF5A (PDB ID 3CPF, ref. 37) as search models. The resulting information and position of selenium atoms were used to obtain the electron density map in AutoSol Wizard[65] in Phenix suite[66]. Model building was carried out with resolve and buccaneer[67] using AutoBuild Wizard[68] in Phenix and with COOT (ref. 69). Iterative cycles of refinement using PHENIX Refine[70] were done after each round of model building and the quality of the model was assessed with MolProbity[71]. In the final stages, the model was refined against a native data set at a resolution of 3.2 Å to an $R_{work}$ of 21.5% and $R_{free}$ of 26.8%. The model has good stereochemistry, with 96% of the residues in the most favoured region of the Ramachandran plot and only 0.2% outliers.

All figures were prepared using USCF Chimera (http://www.cgl.ucsf.edu/chimera).

**Nuclear export assays.** HeLa cell nuclei were prepared as described[23]. Briefly, HeLa cells (originally obtained from ATCC) were grown to ≈95% confluency, washed with PBS + 0.2% glucose, detached with a citrate/EDTA solution, sedimented, and washed 3x with ice-cold transport buffer (20 mM HEPES/KOH pH 7.5, 110 mM KOAc, 5 mM Mg(OAc)$_2$, 0.5 mM EGTA and 250 mM sucrose). Cell membranes were permeabilized with 25 μg ml$^{-1}$ high purity digitonin (in ice-cold transport buffer) and nuclei recovered by centrifugation. After two washes, nuclei were resuspended in transport buffer (without sucrose) containing 5% (w/v) glycerol, 0.5 M trehalose and slowly frozen at − 80 °C. The permeabilization and washing steps deplete endogenous NTRs (such as Xpo4) from the nuclei.

We added *Xenopus laevis* egg extract[72] for nuclear export assays, as it stabilizes the nuclei over hours, but prior depleted all NTRs from the extract by the Phenyl Sepharose method[11,73]. Nuclei (6 μl) were mixed with 54 μl NTR-depleted egg

extract (supplemented with 1 mM ATP, 0.5 mM GTP, 10 mM creatine phosphate, 0.3 μM NTF2 and 3 μM Ran) and incubated for 15 min at 22 °C before addition of 2 μM labelled (hypusinated) eIF5A. Imaging was performed through the 'life samples' with a Leica SP5 confocal laser-scanning microscope, using the 561-nm laser line and a × 63 HCX Pl apo lambda blue 1.4 oil objective (Leica).

**Data availability.** The coordinates and structure factors have been deposited in the Protein Data Bank with accession code 5DLQ. All other data supporting the findings of this study are available within the article and its Supplementary Information file, or from the corresponding author on reasonable request.

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

## Acknowledgements

We thank Gabi Hawlitschek for excellent technical help, Heinz-Jürgen Dehne for the permeabilized HeLa cells, Jens Krull for *Xenopus* egg extract, Bernard Freytag for Ran-energy-regenerating system, Samir Karaca for mass spectrometry analysis, the staff of synchrotron beamlines at the Swiss Light Source (SLS, Villigen, X10SA, PXII) for assistance during data collection, Rüstem Yilmaz, Cornelia Paz, Tino Pleiner and Kevser Fünfgeld for critical reading of the manuscript as well as the Max-Planck-Gesellschaft and the Deutsche Forschungsgemeinschaft (SFB 860) for funding.

## Author contributions

M.A. conceived and performed the experiments, crystallized the export complex, contributed to the crystallographic data collection, performed model building and structure refinement, interpreted the data, prepared figures and wrote the manuscript; S.T. performed the crystallographic data collection, data processing and initial heavy atom identification; D.G. supervised the project, conceived the experiments, interpreted the data and wrote the manuscript.

## Additional information

**Competing financial interests:** The authors declare no competing financial interests.

