## [Peer Review File · Nature Communications]

Reviewer #1 (Remarks to the Author):

This is an interesting and well-written paper describing the structure of Exp4 bound to RanGTP and eIF5A. This work not only expands the structural repertoire of importin β -like receptors for which a 3D structure is available, but also reveals a novel mode of intramolecular NES-mimicry. Though the authors haven't emphasized this point much, personally, I find it the coolest take-home message of this paper.

Overall, the paper is clearly Nature Communications quality.

I have a few suggestions that I invite the authors to address:

1. Fig 7c should go after Fig 3 or become a Suppl Figure, right now the fig order is Fig 1, 2, 3, 7c, 4, etc, which is inconsistent;
2. Crystallographic table: all numbers should consistently have 1 or 2 decimal digits (right now cell axes have 3, angles zero and resolution 2)
3. Crystallographic table: If only 1 crystal was used for data collection, Rmerge should be referred to as Rsym. Alternatively, the authors should spell out how many crystals were used.
4. The Rfree is a bit high for a 3.2Å structure, as well as the Rwork is somewhat uncoupled and low (considering that no solvent was modelled). I wonder if the authors have generated anomalous restraints from the 56 SeMet sites used for phasing. Such a trick per se could lower the Rfree by a point or so.

Reviewer #2 (Remarks to the Author):

This manuscript describes the crystal structure of exportin Xpo4 in complex with eIF5A and RanGTP. The overall data demonstrate that the Xpo4 structure in this complex is very similar to the cargo-bound conformation of Xpo1 and that the subcomplex Xpo4-RanGTP is very similar to other complexes of RanGTP and nuclear transport receptors. On the other hand, differently from cargo recognition by Xpo1, via the short nuclear export sequence (NES), Xpo4 binds extensively to both

domains of eIF5A, including its N-terminal positively charged long loop, which contains the hypusine modification. Finally, the authors confirm the relevance of specific residues in Xpo4 for binding to eIF5A. The manuscript is original and very relevant to understanding the functional interplay between Xpo4 and eIF5A. The methodology and data analysis are adequate and the results are presented clearly.

Suggested improvements:

1- End of Page 3 and beginning of Page 4. A reference from literature should be added about the action of exportins to avoid the leakage of cytoplasmic components.

2- Page 4. A reference should also be added to the end of the phrase "Earlier literature assigned a multitude of other functions to eIF5A; retrospectively however, it is reasonable to assume that these are secondary to the requirement of eIF5A for efficiently translating specific mRNAs".

3- Page 10. Concerning the text "The recognition of very conserved eIF5A features is also (...)": is this unpublished data? It should be clarified.

4- Acetylation of eIF5A residue K47 has been shown to both inhibit eIF5A function in translation (Lee et al., 2011) and to increase its nuclear localization (Ishfaq et al., 2012a; Ishfaq et al., 2012b). This residue is present in the N-terminal positively charged long loop of eIF5A and it also seems to be involved with Xpo4 binding (Figure 5a). What interactions does this residue have with Xpo4 and how could K47 acetylation interfere with them?

5- Although the authors propose the idea that "off-target interactions (of eIF5A) might disturb the assembly process of ribosomal subunits", no data is presented to support this hypothesis. The Xpo4-D470N mutant, for example, loses interaction with eIF5A (Figure 7a) and induces eIF5A nucleolar accumulation (Figure 7c). A polysome profile analysis and analysis of total 40S and 60S subunits in Xpo4-D470N would improve significantly the impact of this manuscript by giving solid support to the reason why Xpo4 is necessary to exclude eIF5A from the nucleus.

6- End of Page 11 and beginning of Page 12. Concerning the discussion about why, during evolution, eIF5A did not increase in size or fuse to some "cytoplasmic anchor" to slow down its passive nuclear entry. It is suggested that the docking site of eIF5A in the ribosome would prevent that. This discussion could be further elaborated taking into account the cryo-electron microscopy reconstruction of eIF5A bound to the 80S ribosome (Schmidt et al., 2016). Also, concerning the last phrase of this paragraph, it should be considered that N-terminal fusions of eIF5A with GFP (Valentini et al., 2002), GST (Zanelli et al., 2006), TAP (Jao & Chen, 2006) and 6xHis (Gutierrez et al., 2013) tags are functionally indistinguishable in comparison with untagged eIF5A. This information does not support the idea of a "fusion-intolerance" of eIF5A and it is unlikely that a short (9-15 residues long) NES peptide would abolish eIF5A function in translation. This paragraph should be considerably improved.

References

Gutierrez E, Shin BS, Woolstenhulme CJ, Kim JR, Saini P, Buskirk AR, Dever TE. eIF5A promotes translation of polyproline motifs. *Mol Cell*. 2013 Jul 11;51(1):35-45. doi: 10.1016/j.molcel.2013.04.021. Epub 2013 May 30. PMID: 23727016

Ishfaq M, Maeta K, Maeda S, Natsume T, Ito A, Yoshida M. The role of acetylation in the subcellular localization of an oncogenic isoform of translation factor eIF5A. *Biosci Biotechnol Biochem*. 2012;76(11):2165-7. Epub 2012 Nov 7. PMID: 23132580

Ishfaq M, Maeta K, Maeda S, Natsume T, Ito A, Yoshida M. Acetylation regulates subcellular localization of eukaryotic translation initiation factor 5A (eIF5A). *FEBS Lett*. 2012 Sep 21;586(19):3236-41. doi: 10.1016/j.febslet.2012.06.042. Epub 2012 Jul 4. Erratum in: *FEBS Lett*. 2014 Aug 19;588(16):2754. PMID: 22771473

Jao DL, Chen KY. Tandem affinity purification revealed the hypusine-dependent binding of eukaryotic initiation factor 5A to the translating 80S ribosomal complex. *J Cell Biochem*. 2006 Feb 15;97(3):583-98. PMID: 16215987

Lee SB, Park JH, Folk JE, Deck JA, Pegg AE, Sokabe M, Fraser CS, Park MH. Inactivation of eukaryotic initiation factor 5A (eIF5A) by specific acetylation of its hypusine residue by spermidine/spermine acetyltransferase 1 (SSAT1). *Biochem J*. 2011 Jan 1;433(1):205-13. doi: 10.1042/BJ20101322. PMID: 20942800

Schmidt C, Becker T, Heuer A, Braunger K, Shanmuganathan V, Pech M, Berninghausen O, Wilson DN, Beckmann R. Structure of the hypusinylated eukaryotic translation factor eIF-5A bound to the ribosome. *Nucleic Acids Res*. 2016 Feb 29;44(4):1944-51. doi: 10.1093/nar/gkv1517. Epub 2015 Dec 28. PMID: 26715760

Valentini SR, Casolari JM, Oliveira CC, Silver PA, McBride AE. Genetic interactions of yeast eukaryotic translation initiation factor 5A (eIF5A) reveal connections to poly(A)-binding protein and protein kinase C signaling. *Genetics*. 2002 Feb;160(2):393-405. PMID: 11861547

Zanelli CF, Maragno AL, Gregio AP, Komili S, Pandolfi JR, Mestriner CA, Lustrri WR, Valentini SR. eIF5A binds to translational machinery components and affects translation in yeast. *Biochem Biophys Res Commun*. 2006 Oct 6;348(4):1358-66. Epub 2006 Aug 7. PMID: 16914118

Reviewer #3 (Remarks to the Author):

The manuscript describes the crystal structure of Xpo4 in complex with RanGTP and eIF5A (hypusine form) and proposes how Xpo4 binding of eIF5A is responsible for nuclear export of eIF5A and how Xpo4 binding shields eIF5A from off-target interactions in nuclei. This is a very interesting study that

illustrates specific molecular interactions between Xpo4, RanGTP and hypusinated eIF5A and provides new insights into the role of Xpo4 as a nuclear transport receptor for other cargos. The experiments were carefully designed and performed with impressive precision and the data, in general, supports the conclusions drawn.

Minor points:

1. On p13-14, there is no description of how recombinant DHS and DOHH enzymes were obtained. Indicate the specific information on their production.
2. Instead of the in vitro DHS/DOHH reaction, the authors could have used a bacterial polycistronic system to overexpress eIF5A, DHS and DOHH and to produce recombinant hypusinated eIF5A from E.coli (PEDS, 2011 24:301-309).
3. How were eIF5A(Lys), eIF5A(Dhp) and eIF5A(Hpu) forms confirmed, other than their resolution from cationic resin and by removal of deoxyhypusine form by DHS reversal reaction? These forms can be validated by using a hypusine specific antibody, amino acid analysis or mass spectrometry.
4. Fig.1b right side panel SDS gel, it is not clear which sample is in each lane. Are they different fractions from Superdex200? What does 'SEC of trypsin-treated complex' mean, as there are no numbers under this label?
5. In Fig. 7C, does WT mean K50(Lys) or K50(Hypusine)? If it is hypusinated eIF5A, it would be also important to show that the ternary complex formation depends on the hypusine modification by testing WT(K50Lys) in this assay.
6. Previously, it was reported that nonhypusinated eIF5A(Lys) (upon transfection of GFP-eIF5A vector without DHS +DOHH vectors) is localized in nuclei/whole cell and hypusinated eIF5A (upon transfection of GFP-eIF5A vector with DHS +DOHH vectors) in cytoplasm in mammalian cells (FEBS Lett 586 (2012), 3236-3241). Nuclear localization of the nonhypusinated or hypusinated eIF5A was attributed to acetylation at Ly47 by PCAF and this acetylation may inhibit XPO4 binding and thereby retain it in nuclei. Therefore, it would be interesting to compare eIF5A variants, eIF5A(Hpu50), eIF5A(Lys50), eIF5A(Lys50, AcLys47) and eIF5A(Hpu50, AcLys47) in the ternary complex formation (Fig. 7C) and nuclear export assay (Fig. 7B).
7. On p4, "It occurs not only in eukaryotes....but also in eubacteria, where it is called EF-P" needs to be changed to "It occurs in eukaryotes and archaea, but not in eubacteria, which contain an eIF5A ortholog, EF-P that does not undergo hypusine modification. "

Point-by-point reply (The respective reviewers' comments are cited in blue, followed by our answers)

Reviewer #1 (Remarks to the Author):

This is an interesting and well-written paper describing the structure of Exp4 bound to RanGTP and eIF5A. This work not only expands the structural repertoire of importin b-like receptors for which a 3D structure is available, but also reveals a novel mode of intramolecular NES-mimicry. Though the authors haven't emphasized this point much, personally, I find it the coolest take-home message of this paper.

Overall, the paper is clearly Nature Communications quality.

Thank you!

I have a few suggestions that I invite the authors to address:

1. Fig 7c should go after Fig 3 or become a Suppl Figure, right now the fig order is Fig 1, 2, 3, 7c, 4, etc, which is inconsistent;

We mention Figure 7c early in the text to document the observation that abolishing the single salt bridge between Ran and eIF5A has no effect on the interactions. Figure 7 combines all mutagenesis data, with the different mutations serving as mutual controls. This coherence in Figure 7 would be lost if we disintegrated the figure into smaller pieces. On the other hand, we take the point that the figure order is not perfectly consistent. We solved this problem by changing the reference to the figure to read "see below, Fig. 7c".

2. Crystallographic table: all numbers should consistently have 1 or 2 decimal digits (right now cell axes have 3, angles zero and resolution 2)

We changed all numbers to have two decimal digits.

3. Crystallographic table: If only 1 crystal was used for data collection, Rmerge should be referred to as Rsym. Alternatively, the authors should spell out how many crystals were used.

We used one crystal per condition. We changed the table accordingly.

4. The Rfree is a bit high for a 3.2Å structure, as well as the Rwork is somewhat uncoupled and low (considering that no solvent was modelled). I wonder if the authors have generated anomalous restraints from the 56 SeMet sites used for phasing. Such a trick per se could lower the Rfree by a point or so.

Exportins in general are not only difficult to crystallise, but also often contain regions of considerable flexibility within the crystals. This was a major problem we faced during model building. Yet, we have improved the structure considerably and finally arrived at an Rfree of 26.8% (down from 29.9%) and Rfree-Rwork of 5.3% (down from 6.3%), with still excellent geometry.

Changes in the refinement procedure were:

- We added another 58 residues to the model
- We modelled another 459 side chains in regions that previously had only been built as polyalanine or with partial side chains (mainly in the C-terminal regions of the two Xpo4 molecules in the asymmetric unit).
- We applied a TLS refinement to distinct regions of the complexes

The overall difference to the previous structure is small (r.m.s.d. of 0.1 Å for the backbones), but there are improvements in some flexible regions. There is no discernable structural change in the eIF5a•Xpo4 and Ran•Xpo4 interfaces. Thus, all previously drawn conclusions about the modes of interactions are also fully supported by the improved structure.

As can be seen from the Table, our structure in terms of Rfree is now one of the best export complex structures in the PDB.

Structure	PDB	Resolution (Å)	Rwork (%)	Rfree (%)
CAS•Ran•Impα	1WA5	2.00	23.4	26.8
Xpo4•Ran•eIF5A	5DLQ	3.20	21.5	26.8
Crn1•Ran•Spn1	3GJX	2.50	24.4	28.1
Crn1•Ran•HIV1-NES	3NBZ	2.80	22.6	28.5
Imp13•Mago•Y14	2X1G	3.35	26.7	28.9
Xpot•Ran•tRNA	3ICQ	3.20	24.1	29.4
Imp13•Ran•eIF1A	3ZJY	3.60	26.8	29.9
Xpo5•Ran•miRNA	3A6P	2.90	24.7	31.2
Crn1•Ran•PKI-NES	3NBY	3.42	25.8	31.5

Reviewer #2 (Remarks to the Author):

This manuscript describes the crystal structure of exportin Xpo4 in complex with eIF5A and RanGTP. The overall data demonstrate that the Xpo4 structure in this complex is very similar to the cargo-bound conformation of Xpo1 and that the subcomplex Xpo4-RanGTP is very similar to other complexes of RanGTP and nuclear transport receptors. On the other hand, differently from cargo recognition by Xpo1, via the short nuclear export sequence (NES), Xpo4 binds extensively to both domains of eIF5A, including its N-terminal positively charged long loop, which contains the hypusine modification. Finally, the authors confirm the relevance of specific residues in Xpo4 for binding to eIF5A. The manuscript is original and very relevant to understanding the functional interplay between Xpo4 and eIF5A. The methodology and data analysis are adequate and the results are presented clearly.

Thank you!

Suggested improvements:

1- End of Page 3 and beginning of Page 4. A reference from literature should be added about the action of exportins to avoid the leakage of cytoplasmic components.

We added references for exportins retrieving actin and translation factors back to the cytoplasm. This takes us beyond the limit of 70 references, but we entirely agree that this statement needed a citation.

2- Page 4. A reference should also be added to the end of the phrase "Earlier literature assigned a multitude of other functions to eIF5A; retrospectively however, it is reasonable to assume that these are secondary to the requirement of eIF5A for efficiently translating specific mRNAs".

The ending of the paragraph was actually meant to refer to its beginning, namely the fact that eIF5A is required for synthesising proteins containing consecutive proline residues. We re-phrased this as follows: "*Earlier literature assigned a multitude of other functions to eIF5A; retrospectively however, it is reasonable to assume that these are secondary to the requirement of eIF5A for efficiently translating mRNAs that encode proteins with proline-rich motifs.*" The last-mentioned fact is referenced already in the preceding sentence.

3- Page 10. Concerning the text "The recognition of very conserved eIF5A features is also (...)": is this unpublished data? It should be clarified.

We now write: "*The recognition of very conserved eIF5A features is also evident from our observation that mammalian Xpo4 binds yeast (*S. cerevisiae*) eIF5A perfectly well (M.A., A. Rodriguez, D.G., unpublished data)*".

4- Acetylation of eIF5A residue K47 has been shown to both inhibit eIF5A function in translation (Lee *et al.*, 2011) and to increase its nuclear localization (Ishfaq *et al.*, 2012a; Ishfaq *et al.*, 2012b). This residue is present in the N-terminal positively charged long loop of eIF5A and it also seems to be involved with Xpo4 binding (Figure 5a). What interactions does this residue have with Xpo4 and how could K47 acetylation interfere with them?

Lee *et al.*, 2011 discuss acetylation of the hypusine (= K50), which of course will deteriorate Xpo4 binding. The same authors published earlier (Lee *et al.*, 2009) that mutations of K47 to either alanine or arginine had no effect on localisation. This is fully consistent with our structure, where we also see no evidence for K47 contributing to Xpo4-binding. The closest contact from Xpo4 is the carboxylate of Asp698 at a distance of 5.2Å, which appears too far to be relevant. Acetylation of K47 should produce no clashes and should therefore be compatible with recognition by the exportin.

Hypusination blocks acetylation at K47 (Lee *et al.*, 2009, 2011; Ishfaq *et al.*, 2012). In fact, endogenous K47-acetylated eIF5A is essentially non-detectable in cells unless hypusination is inhibited by the small molecule inhibitor GC7 (Ishfaq *et al.*, 2012) or eIF5A over-expression overwhelms the hypusination system. To us, this suggests that K47 acetylation is of little relevance for localising the physiologically active form of eIF5A, the actual subject of our study. We therefore prefer not to distract the reader by discussing K47. Instead, we think that this would be more appropriate for a review.

5- Although the authors propose the idea that "off-target interactions (of eIF5A) might disturb the assembly process of ribosomal subunits", no data is presented to support this hypothesis. The Xpo4-D470N mutant, for example, loses interaction with eIF5A (Figure 7a) and induces eIF5A nucleolar accumulation (Figure 7c). A polysome profile analysis and analysis of total 40S and 60S subunits in Xpo4-D470N would improve significantly the impact of this manuscript by giving solid support to the reason why Xpo4 is necessary to exclude eIF5A from the nucleus.

This issue is actually a bit more complicated. The D470N mutation is a loss-of-function-mutation. Therefore, one cannot just transfect the mutant and expect a phenotype. Instead, all wild type alleles need to be either inactivated or replaced by the mutant one. To do this in a genetically clean way with all necessary controls and validations would be a major undertaking, even when using the CRISP/Cas9 system (note that we are working with the mouse exportin).

In addition, Xpo4 exports not only eIF5A but carries numerous additional cargoes into and out of the nucleus (our latest mass spec dataset suggests at least 50). For a meaningful interpretation of the suggested experiment, we would need to explore how the mutation affects other cargoes (either directly or indirectly through a loss of competition or synergy with eIF5A export), and we probably would have to disentangle the impacts of multiple transport effects on translation. Considering that our manuscript focuses on the structure of the export complex, this would take us rather far off our topic.

6- End of Page 11 and beginning of Page 12. Concerning the discussion about why, during evolution, eIF5A did not increase in size or fuse to some "cytoplasmic anchor" to slow down its passive nuclear entry. It is suggested that the docking site of eIF5A in the ribosome would prevent that. This discussion could be further elaborated taking into account the cryo-electron microscopy reconstruction of eIF5A bound to the 80S ribosome (Schmidt et al., 2016). Also, concerning the last phrase of this paragraph, it should be considered that N-terminal fusions of eIF5A with GFP (Valentini et al., 2002), GST (Zanelli et al., 2006), TAP (Jao & Chen, 2006) and 6xHis (Gutierrez et al., 2013) tags are functionally indistinguishable in comparison with untagged eIF5A. This information does not support the idea of a "fusion-intolerance" of eIF5A and it is unlikely that a short (9-15 residues long) NES peptide would abolish eIF5A function in translation. This paragraph should be considerably improved.

We now discuss the Schmidt 2016 paper and we re-phrased the paragraphs as follows: "*eIF5A has an invariably small molecular mass of only 17 kDa and can thus cross the permeability barrier of NPC rather rapidly even without the help of any importin. An increase in size would have been a straightforward way of slowing down such passive nuclear entry. Yet, a fusion to a sufficiently large 'cytoplasmic anchor' was not an evolutionary successful solution, perhaps because the present (small-sized) forms of eIF5A represent the optimum for rapid and accurate translation and because even slight reductions in translation efficiency (that might come with an increased eIF5A mass) would decrease the competitive fitness of the organism.*"

Reviewer #3 (Remarks to the Author):

The manuscript describes the crystal structure of Xpo4 in complex with RanGTP and eIF5A (hypusine form) and proposes how Xpo4 binding of eIF5A is responsible for nuclear export of eIF5A and how Xpo4 binding shields eIF5A from off-target interactions in nuclei. This is a very interesting study that illustrates specific molecular interactions between Xpo4, RanGTP and hypusinated eIF5A and provides new insights into the role of Xpo4 as a nuclear transport receptor for other cargos. The experiments were carefully designed and performed with impressive precision and the data, in general, supports the conclusions drawn.

Thank you!

Minor points:

1. On p13-14, there is no description of how recombinant DHS and DOHH enzymes were obtained. Indicate the specific information on their production.

We added the requested information to the Methods: "*Deoxyhypusine synthase and deoxyhypusine hydroxylase were expressed as N-terminal His10-GFP-Tev and His21-Tev fusions, respectively, as described for eIF5A variants. The two enzymes were purified by Ni(II) chelate chromatography and used in tagged form, so that they could be removed after the reaction.*"

2. Instead of the in vitro DHS/DOHH reaction, the authors could have used a bacterial polycistronic system to overexpress eIF5A, DHS and DOHH and to produce recombinant hypusinated eIF5A from E.coli (PEDS, 2011 24:301-309).

That is true. However, the enzymatic modification was easier for us to implement and worked at the very first attempt.

3. How were eIF5A(Lys), eIF5A(Dhp) and eIF5A(Hpu) forms confirmed, other than their resolution from cationic resin and by removal of deoxyhypusine form by DHS reversal reaction? These forms can be validated by using a hypusine specific antibody, amino acid analysis or mass spectrometry.

Our crystal structure is probably the best answer to the question: we see clear electron density for hypusine, including that of the hydroxyl group.

At a very early stage of the project, we tried mass spectrometry. Upon modification, we saw the expected loss of tryptic peptides, but the setup at our facility was at that time not ideal for a positive identification of the hypusine-containing peptide. We therefore relied on the above-mentioned criteria as well as on the strength of Xpo4-binding as readouts.

4. Fig.1b right side panel SDS gel, it is not clear which sample is in each lane. Are they different fractions from Superdex200? What does 'SEC of trypsin-treated complex' mean, as there are no numbers under this label?

We now clarified the lettering to read: "*Superdex200 fractions (10-13 ml) of trypsin-treated complex*".

5. In Fig. 7C, does WT mean K50(Lys) or K50(Hypusine)? If it is hypusinated eIF5A, it would be also important to show that the ternary complex formation depends on the hypusine modification by testing WT(K50Lys) in this assay.

All eIF5A variants tested in this panel were K50(Lys), otherwise they would not have been comparable. To compensate for the reduced affinity (without hypusination), the binding reaction was performed at lower salt. We changed the word order in the legend to make this clearer.

6. Previously, it was reported that nonhypusinated eIF5A(Lys) (upon transfection of GFP-eIF5A vector without DHS +DOHH vectors) is localized in nuclei/whole cell and hypusinated eIF5A (upon transfection of GFP-eIF5A vector with DHS +DOHH vectors) in cytoplasm in mammalian cells (FEBS Lett 586 (2012), 3236-3241). Nuclear localization of the nonhypusinated or hypusinated eIF5A was attributed to acetylation at Ly47 by PCAF and this acetylation may inhibit XPO4 binding and thereby retain it in nuclei. Therefore, it would be interesting to compare eIF5A variants, eIF5A(Hpu50), eIF5A(Lys50), eIF5A(Lys50, AcLys47) and eIF5A(Hpu50, AcLys47) in the ternary complex formation (Fig. 7C) and nuclear export assay (Fig. 7B).

The quoted paper (Ishfaq *et al.*, 2012) actually does not provide any support to the idea that acetylation at K47 would cause nuclear accumulation of hypusinated eIF5A. The gap in the arguments is that the fluorescent images of the transfected cells do not distinguish between hypusinated, deoxyhypusinated, and non-hypusinated forms (which for sure co-existed in their experimental setup). In fact, their data strongly suggest that hypusinated eIF5A is not K47-acetylated and *vice versa*.

As mentioned in our answer to referee 2, our structure is neither consistent with a direct contribution of K47 to Xpo4-binding nor does it suggest that acetylation at K47 produces any clashes. In that, we agree with Lee *et al.*, 2009, who found that K47A and K47R mutations had no effect on the eIF5A localisation. Specifically, they wrote: "Lys47 acetylation does not seem to be a major factor regulating eIF5A subcellular localization".

Already 16 years ago we published that a loss of hypusination causes a 40-fold drop in affinity for Xpo4 (see Table I in Lipowsky *et al.*, 2000) and this was subsequently confirmed by others (see e.g. (Lee *et al.*, 2009)). Perhaps there is no need to show such experiment again. It is also obvious that the (non-hypusinated) eIF5A(AcLys47) variant will only poorly bind to Xpo4.

eIF5A(Hyp, AcLys47): We have not yet seen any evidence that this double-modified variant exists in cells, or if there is a practical way of producing it. PCAF does not acetylate hypusinated eIF5A, and it is unclear if the hypusinating enzymes DHS and DOHH would accept eIF5A(AcLys47) as a substrate. Considering that our manuscript focuses on the crystal structure of the eIF5A•Xpo4•RanGTP complex, we also feel that this is not quite our topic. Instead, such discussion would better fit into a review.

7. On p4, "It occurs not only in eukaryotes....but also in eubacteria, where it is called EF-P" needs to be changed to "It occurs in eukaryotes and archaea, but not in eubacteria, which contain an eIF5A ortholog, EF-P that does not undergo hypusine modification. "

The suggested phrasing does not quite fit our introduction, because hypusination is described only later in the text, after clarifying homologies and function. Furthermore, if we agree that eIF5A and EF-P are orthologues that carry out the same function, then it is also justified to state that the protein occurs in eukaryotes, archaea and eubacteria.

We agree that we should avoid the impression that eubacterial EF-P is hypusinated. We therefore now begin the introductory paragraph on hypusination with the specification that we are talking about the eukaryotic and archaeobacterial forms.

References

- Ishfaq M, Maeta K, Maeda S, Natsume T, Ito A, Yoshida M. 2012. Acetylation regulates subcellular localization of eukaryotic translation initiation factor 5A (eIF5A). *FEBS Lett*, **586**:3236–3241. doi:10.1016/j.febslet.2012.06.042.
- Lee SB, Park JH, Folk JE, Deck JA, Pegg AE, Sokabe M, Fraser CS, Park MH. 2011. Inactivation of eukaryotic initiation factor 5A (eIF5A) by specific acetylation of its hypusine residue by spermidine/spermine acetyltransferase 1 (SSAT1). *Biochem J*, **433**:205–213. doi:10.1042/BJ20101322.
- Lee SB, Park JH, Kaebel J, Sramkova M, Weigert R, Park MH. 2009. The effect of hypusine modification on the intracellular localization of eIF5A. *Biochem Biophys Res Commun*, **383**:497–502. doi:10.1016/j.bbrc.2009.04.049.
- Lipowsky G, Bischoff FR, Schwarzmaier P, Kraft R, Kostka S, Hartmann E, Kutay U, Görlich D. 2000. Exportin 4: a mediator of a novel nuclear export pathway in higher eukaryotes. *EMBO J*, **19**:4362–4371. doi:10.1093/emboj/19.16.4362.

Reviewer #1 (Remarks to the Author):

I am satisfied by the way the authors addressed all of my comments. Significantly, the re-refined structural model is much improved. I thank and praise the authors for improving the Rfree by 4 points! Overall, this is a very nice paper that fits well in Nature Communications.

Reviewer #2 (Remarks to the Author):

The authors' responses addressed properly most of the comments for suggested improvements. However, concerning the text "off-target interactions (of eIF5A) might disturb the assembly process of ribosomal subunits", this reviewer still thinks this is very speculative and should be removed from the manuscript or more experimental support should be given to this hypothesis. Furthermore, although the authors justify that "one cannot just transfect the mutant and expect a phenotype", a clear nuclear accumulation of eIF5A is observed for transfected Xpo4 mutants in Figure 7b. Therefore, one cannot rule out the possibility of a polysome profile phenotype using the same experiment conditions as in Figure 7b. Also, expression of an NLS-eIF5A or other eIF5A fusions based on Xpo4-eIF5A complex structure could be designed to address this question.

Reviewer #3 (Remarks to the Author):

The authors have addressed all the points raised by the reviewers and the manuscript is acceptable for publication in Nature Communications.

Answers to reviewers' points:

REVIEWERS' COMMENTS:

Reviewer #1 (Remarks to the Author):

I am satisfied by the way the authors addressed all of my comments. Significantly, the re-refined structural model is much improved. I thank and prize the authors for improving the Rfree by 4 points! Overall, this is a very nice paper that fits well in Nature Communications.

Thank you!

Reviewer #2 (Remarks to the Author):

The authors' responses addressed properly most of the comments for suggested improvements. However, concerning the text "off-target interactions (of eIF5A) might disturb the assembly process of ribosomal subunits", this reviewer still thinks this is very speculative and should be removed from the manuscript or more experimental support should be given to this hypothesis. Furthermore, although the authors justify that "one cannot just transfect the mutant and expect a phenotype", a clear nuclear accumulation of eIF5A is observed for transfected Xpo4 mutants in Figure 7b. Therefore, one cannot rule out the possibility of a polysome profile phenotype using the same experiment conditions as in Figure 7b. Also, expression of an NLS-eIF5A or other eIF5A fusions based on Xpo4-eIF5A complex structure could be designed to address this question.

Our previous statement "one cannot just transfect a (loss-of-function) mutant and expect a phenotype" is still absolutely valid. The experiment shown in Figure 7b did not involve any transfections (the term does not appear anywhere in the paper). It is a permeabilized cell assay, as it has been standard in the nuclear transport field for by now nearly three decades.

The assay starts with growing (wild type) HeLa cells, permeabilizing their plasma membrane with digitonin (allowing soluble components to leak out of the cells), and washing them thoroughly. This treatment depletes endogenous nuclear transport receptors (NTRs) to very low levels, thus making active nuclear transport dependent on the re-addition of exogenous NTRs (Xpo4 in our case).

The suggested experiment would only work if the ribosomes that pre-exist within the system (originating from the HeLa cells and the cycloheximide-stabilised interphase *Xenopus* egg extract) would get exchanged for newly assembled ones. This, however, will not happen, because (during the short assay time) there will be no noticeable turnover of the existing human and frog ribosomes, and the conditions for making and exporting new ribosomes are simply not given.

Reviewer #3 (Remarks to the Author):

The authors have addressed all the points raised by the reviewers and the manuscript is acceptable for publication in Nature Communications.

Thank you!